# Effect of Nucleating Additives on Short- and Long-Term Tensile Strength and Residual Stresses of Welded Polypropylene Samples

**DOI:** 10.3390/polym13172965

**Published:** 2021-08-31

**Authors:** Andrea Wübbeke, Volker Schöppner, Theresa Arndt, Jan-Ole Maras, Marcus Fitze, Christian Moltzahn, Tao Wu, Thomas Niendorf

**Affiliations:** 1Kunststofftechnik Paderborn, Universität Paderborn, Warburger Straße 100, 33098 Paderborn, Germany; andreawuebbeke@googlemail.com (A.W.); volker.schoeppner@ktp.uni-paderborn.de (V.S.); 2Theresa Arndt, Kunststofftechnik Paderborn, Universität Paderborn, Warburger Straße 100, 33098 Paderborn, Germany; 3Department Chemie, Universität Paderborn, Warburger Straße 100, 33098 Paderborn, Germany; jmaras@campus.uni-paderborn.de (J.-O.M.); marcus-fitze@gmx.de (M.F.); christian.moltzahn@gmx.de (C.M.); 4Institut für Werkstofftechnik, Metallische Werkstoffe, Universität Kassel, Sophie-Henschel-Haus Mönchebergstraße 3, 34125 Kassel, Germany; niendorf@uni-kassel.de

**Keywords:** short-term tensile strength, long-term tensile strength, residual stresses, polypropylene, nucleating agents, hot plate welding

## Abstract

Additives such as nucleating agents are often used in the processing of plastic products not only for improving production efficiency but also for enhancing mechanical performance. In this work, the short- and long-term tensile strength, the morphology as well as the residual stresses of the welded polypropylene (PP) samples with different fillers (carbon black and special beta-nucleating agents) and different dimensionless joining paths are analyzed. Results obtained are then compared with those that are representative of the initial, filler-free samples. It is shown that, upon using the special beta-nucleating agent, superior long-term tensile strength can be achieved compared to the samples without additives or with carbon black agent (e.g., for the dimensionless joining path of 0.95, the long-term tensile strength of a PP nature sample is characterized by around 400 MPa, whereas by adding beta-nucleating agent 1% the value can be increased by 400% to reach around 2050 MPa). However, adding beta-nucleating agent 1% yields inferior short-term tensile strength. The hole drilling method (HDM) is used for the analysis of residual stresses. It is found that the residual stresses in the weld seam are characterized by low values of the tensile stresses. The residual stresses in the weld seam also can be converted from tensile into compressive stresses by adding the beta nucleating agent. However, this has the disadvantage that with a higher proportion by weight of the beta nucleating agent, the short-term tensile strength of the welded joint becomes lower than that of the other tested bonds.

## 1. Introduction

Polypropylene (PP) is a polymer material with high industrial relevance. It can be used in a wide range of possible applications—e.g., in the automotive sector for applications such as instrument panels, center consoles, ventilation, and headlight housings. The consumption of polypropylene is currently over 50 million tons per year worldwide. This material has high strength and rigidity combined with low density (0.905 to 0.915 g/cm^3^) [1]. Polymer materials are usually formulated with various fillers, reinforcing materials and additives to form compounds, so that the desired properties can be achieved. Moreover, additives are used, for example, to color the material or to support crystallization.

The steady progress in the development of innovative plastic products leads to their increased presence in all segments of life. At the end of the production chain, a central aspect for successful implementation of products is the joinability of a given number of components. Processes considered include welding techniques such as hot plate welding, which has been applied for over 80 years to join plastics by adhesive bonding. For applying this technology, the joining surfaces of the molded parts to be welded are heated by contact or radiation with the aid of a tempered heating element and then are joined under pressure [2]. The cooling conditions lead to the formation of tensile stresses in the weld seam as well as compressive stresses in the base material. These stresses have a negative influence on the mechanical properties and the service life of the component. It is desirable to keep these stresses as low as possible by adding nucleating agents, thus improving the mechanical properties of the component.

The measurement of the residual stresses can be generally classified as non-destructive, semi-destructive, and destructive. Among these methods, the most widely used non-destructive methods for measuring residual stresses in polymer materials are the photoelasticity method and the X-ray diffraction method [3]. However, the photoelasticity method cannot be carried out for numerous polypropylene samples which have no transparency or are opaque. In addition, the photoelastic stress analysis does not provide absolute residual stress values, but only the difference between both principal residual stress components. Thus, it cannot be applied for determining residual stresses in the presence of an equibiaxial residual stress state [3]. The X-ray diffraction method fails for residual stress measurement of non-crystalline materials. Alternatively, the hole drilling method (HDM), as a widely used semi-destructive method, is capable of providing reliable in-depth residual stress results of the plastic samples [4]. Magnier et al. [5] took into account the effects of temperature fluctuations and viscoelastic phenomena occurring upon drilling for evaluating the residual stresses. The reliability of residual stress measurements of plastic materials through HDM has already been validated by mechanical bending tests [4].

Short-term tensile strength is referred to as the maximum stress that a material can withstand before breaking under tensile loading [6]. Long-term tensile strength, also called creep tensile strength, is a property of a material to retain its bearing capacity under a given constant stress during a relatively long time period [7], which affects the lifetime of the samples. In the past few years, many studies have been carried out to investigate the short- and long-term behavior of the bonded plastic samples with different additives. Battisti et al. [8] studied the influence of adding additives to the polypropylene-based polymers on their short- and long-term properties, which were made by the injection molding machine. It was found that the mechanical properties of the investigated materials show a significant dependency on the filler content with an increase of the Young’s modulus up to 48% by adding layered silicates. In [9], the effect of welding parameters on the long-term properties of friction stir-welded polyethylene plates was investigated. It was found that, under special conditions, the creep resistance of the welds is better than that of the base materials.

Polypropylene can be divided into three different spherulite modifications: alpha, beta, and gamma [10,11]. A spherulite describes a globular superstructure, which can be seen using a light microscope [1]. Another well-established technique to visualize the material structure is the wide-angle X-ray scattering (WAXS). With this method, it is possible to distinguish among all three types of spherulite structures based on their intensity pattern [10]. In the industrial context only, the modifications alpha and beta are relevant [12]. The alpha and beta phases differ in their melting temperature, whereby the melting point of alpha (165 °C) is higher than of beta (150 °C) [13,14]. These differences can be distinguished by dynamic differential calorimetry, e.g., with a heating rate of 2 °C/min [13]. Furthermore, both modifications can be distinguished by a polarization microscope on the basis of their birefringence [15,16,17,18]. The microstructures determine macroscopic mechanical properties of polypropylene samples. For example, a high beta component ensures a more favorable energy absorption capability under impact load compared to the alpha modification [14].

In the present work, the polypropylene samples with different additives (beta-nucleating and carbon black agents) are joined together by hot plate welding. The influences of adding different additives on the short- and long-term tensile properties, the morphology as well as the residual stress state of the welded polypropylene samples being processed with different joining paths are experimentally measured and analyzed. To date, there is no research studying long-term tensile strength of the polypropylene samples with speical beta-nucleating agents, which have good impact strength properties. Moreover, the results of short- and long-term tensile strength are compared and a conclusion of their correlation is addressed. Special attention is paid to the variability of the morphology by choosing different additivies and joining paths. The morphological characterization results are used to explain the measurement results of short- and long-term tensile properties of the welded polypropylene samples. In addition, the residual stresses of the welded samples are measured through the hole drilling method (HDM), taking into account the effect of thermal fluctuations and viscoelastic phenomena, for improving the reliability of the measurement results. Changing the additives and joining paths could lead to the variation not only in the absolute value of the residual stress but also in their sign (e.g., switch from tensile to compressive stresses). It is well known that the residual stresses play an important role in long-term mechanical performance of the samples. Based on the measurement and analysis results from the present paper, the objective to establish a deep understanding of the additives/joining path-morphology-mechanical performance (short- and long-term tensile strength) of the welded polypropylene samples can be achieved. Thus, for further investigations the associated process parameters and additives can be optimized to obtain an optimal mechanical performance of the welded polypropylene samples.

## 2. Experiments

### 2.1. Materials

In the present study, Moplen HP501L (LyondellBasell, molecular weight: 342,748 g/mol) was used as the starting polypropylene material. Different additives including beta-nucleating and carbon black agents are added to the polypropylene material for improving mechanical performance. The mixing of the additives with the base material was done manually for preventing the molecular chains from breaking down. The properties of polypropylene depend on its molecular weight. Compounding in a twin screw was not performed upstream as it would lead to a shortening of the molecular chain. A particle size distribution (c.f. Figure 1) of the used carbon black was employed. Since these additives involve high costs, it is advantageous for the industry if a small quantity is sufficient. Based on this, the weight percentages of 0.1% and 1% were selected to investigate whether a small amount of the additives is sufficient to positively modify the mechanical properties. Due to its nucleating properties and cost-effective procurement, carbon black is frequently used in industry. The special beta nucleating agents, on the other hand, are often used because of their good impact strength properties. In combination with the base polypropylene, five different materials are fabricated for analysis, see Table 1, showing the type and weight percentage of additives, material identification used in this work, and structure phases.

The finished mixture was then added to an injection molding machine for producing the polypropylene samples. The test samples have a thickness of 4 mm and are characterized by the shape of half a dog bone, see Figure 2a. The fabricated polypropylene samples with different additives are bonded using hot plate welding. For hot plate welding, a heating element temperature of 220 °C was selected for the experimental investigations. The initial melt layer thickness Lo was measured for all materials according to [19] and the measured Lo revealed a value of 1.2 mm for all modifications. The dimensionless joining path is defined as the ratio of joining path sf (c.f. Figure 2b) to initial melt layer thickness Lo. Different dimensionless joining paths resulted from a change of the weld seam dimensions. Generally, a higher dimensionless joining path was induced by a higher melt fraction. All relevant process parameters are shown in Table 2. In addition, previous investigations have shown that the welding process has no influence on the crystallinity of the material [19,20].

### 2.2. Determination of Residual Stress by the Hole Drilling Method (HDM)

In the case of the HDM, a very small hole is drilled incrementally at the geometrical center of a strain gauge rosette, which is glued to the surface of the part being examined (see Figure 3). The strains released upon drilling are measured. After the layer-wise removal of material, a new equilibrium is established each time around the hole by releasing the residual stresses. A widely accepted method to calculate residual stresses from the strain relaxations is the integral method. For determining the calibration coefficients in the integral method, finite element analysis is required. For more information on the employed theory in HDM and the method of calculating the calibration coefficients, the readers can refer to [21].

The strain gauges used for residual stress analysis were of type Vishay EA-06-062RE-120. As the material under consideration was polypropylene, the surface was treated with a primer before attaching the strain gauge for improving the adhesion between strain gauge and sample. Strain gauges were supplied with a 0.5 V feed voltage to avoid generating too much Joule heat, which is crucial due to the low thermal conductivity of plastic materials. Strain measurements were carried out only 6 h after connecting the strain gauge to the amplifier. This allowed for achieving thermal equilibrium, resulting in a satisfactory stable signal. The drilling process was done manually, with a drilling speed of about 20 rpm and a feed rate of about 0.03 mm/min. As no cooling system was used, this low drilling speed avoids heating when drilling. The drilling tool had a diameter of 2 mm and was custom made flat to ensure a cylindrical geometry of the blind hole without chamfer. In the work, the effect of thermal fluctuations and viscoelastic phenomena upon drilling on the strains was accounted for according to the proposed procedure in [3,5]. Figure 4a highlights the most relevant measuring direction (X-direction) for the samples while Figure 4b shows one welded sample including an attached strain gauge. For all tested samples, the weld bead (see dotted line) was removed in advance to ensure accessibility for the strain gauge rosette and, eventually, to determine the residual stresses in the weld. The strain gauge rosette was applied in such a way that the X-direction reflects the values in the tensile direction (and the Y-direction, not being considered in the remainder of present study indicates the transverse stresses).

### 2.3. Characterization of Short- and Long-Term Tensile Strength and Morphology

For characterization of the short-term tensile strength of the welded polypropylene samples with additives, the tensile tests according to DIN EN 12814-2 [6] (five samples for each test point) were performed at 20 °C (under standard conditions) with a test speed of 50 mm/min. The long-term tensile test was performed four times for each test point at 60 °C ± 0.1 °C according to DIN EN-12814-3 [7], where the environment for these tests included water in which the wetting agent IGEPAL CO 630 was used at a concentration of 0.2% (in mass-%). The DVS2203-4 guideline was consulted for measuring the long-term tensile strength in the creep tensile test. It states that a test temperature below 100 °C should be selected and the tests should be performed at elevated temperatures. Therefore, it was decided to perform the tests at 60 °C. With respect the morphological analysis, specimens were cut out of the welded parts and prepared by grinding and polishing for a permanganate etching with sulfuric acid, phosphoric acid, and potassium permanganate. This method is used to remove the amorphous areas of polyolefin polymers to make crystalline structures visible. To analyze the morphology, the diameters of the α-spherulites were measured with a confocal laser scanning microscope Olympus LEXT OL3100 (Olympus, Tokyo 163-0914, Japan). The WAXS analysis was performed for all welds. A bruker WAXS diffractometer was used in reflection mode in Bragg–Bentano configuration using Cu K (alpha) radiation as light source. The samples were rotated during the measurement.

## 3. Results and Discussion

### 3.1. Morphological Characterization

The microstructures of the samples without any additive with four different dimensionless joining paths (0.3, 0.5, 0.75, 0.95) are illustrated in Figure 5a–d. The WAXS measurement results for different joining paths are shown in Figure 5e. These investigations were carried out on the area (an ellipse with blue color) of the samples marked in Figure 4b. In the untreated samples the α structure is mainly formed. This is also evident in the WAXS measurements shown in Figure 5e, where the characteristic pattern of α structure is present. Occasionally, the shear in the weld bead leads to the formation of β-spherulites. These shine brighter than the α-spherulites due to their birefringence. With respect to the dimensionless joining path of 0.95, it is shown that β-spherulites are found in the recrystallized zone at the boundary between the weld bead and the base material. This phenomenon shows that there is precipitation of β-spherulites due to the shear forces that occur. They are also produced by the shearing of the welding process. An example of the distinction between α and β-spherulites is given in the following Figures, which include an enlargement. This leads to a reduction of the recrystallized zone. As additives, carbon black and special beta nucleating agents (MP2000, from Mayzo) [23] with the weight percentages 0.1% and 1% were used in order to compare the properties of different microstructures (see Table 1).

Figure 6 shows the microstructures and WAXS measurement results of the welded PP carbon black 0.1, accounting for different dimensionless joining paths. This material combination also shows that the characteristic pattern for α structure in the WAXS measurements. The use of carbon black leads to nucleation. Therefore, the formation of the β-spherulites is clearly visible as a brighter shining structure. It becomes clear that in comparison to untreated polypropylene much more β structure occurs. Also, there is a reduction of the recrystallized zone. The densification of the β-spherulites in the joining zone due to the increased joining path is evident in Figure 6b. Increased joining path results in an area at the edge of the joining zone where the stretched α-spherulites form (c.f. Figure 6c,d).

The thin sections of the polypropylene samples with 1% carbon black (c.f. Figure 7), show the strong formation of the β-spherulites although the WAXS measurements show the pattern for an α structure here. These appear even more clearly than the samples with a weight percentage of 0.1% carbon black. This is due to the increased nucleation by the carbon black material.

Figure 8 shows the thin sections of the polypropylene sample with 0.1% weight percentage of β nucleating agent. It can be observed that the β structure predominates. This structure can also be recognized by the light yellowish coloration. These microscopic results are supported by the WAXS measurements (c.f. Figure 8e). Compared to the material combinations considered so far, this material combination exhibits the pattern characteristic of β structure with a peek at 8 degrees, see Figure 8e. It is noticeable that the α-structure is seen exclusively in the boundary layer of the recrystallized zone. This can be clearly seen by the different birefringence. However, the recrystallized zone is reduced again with increasing dimensionless joining path. In Figure 8c, it can be seen that this α-boundary layer has sheared α-spherulites, which can also be seen in Figure 8d. The same is also evident for the β nucleated samples with a weight percentage of 1% (c.f. Figure 9). An α-boundary layer can be seen at the edge of the joining zone. The boundary layer, which was clearly visible in the previous examples PP beta 0.1%, can be seen here only sporadically.

In the samples for polypropylene natural and carbon black, holes are formed in the recrystallized zone, mostly for the joining paths 0.3 and 0.5, due to the low joining pressure. At the same time, however, it can be seen that this phenomenon does not occur for the materials nucleated with β.

### 3.2. Short-Term Tensile Strength

In order to determine the effect of the additives on the short- and long-term tensile strength as well as the residual stresses, the basic polypropylene was also studied for reference. Figure 10 shows the measured short-term tensile strength of the tested materials (PP nature, PP carbon black 0.1, carbon black 1, PP beta 0.1, and PP beta 1) with different dimensionless joining paths (0.3, 0.5, 0.75, and 0.95). It is clearly seen that for all tested materials, as the dimensionless joining path increases, the short-term tensile strength increases and then a plateau is formed after the joining path of 0.5. This can be explained by the fact that, as the dimensionless joining paths increases, the weld seam dimensions also increase. In such cases, the interlocking effect can be improved, thus increasing the short-term tensile strength of the samples. For the dimensionless joining path of 0.3, on the other hand, poor mechanical properties characterized by low strength values are revealed. For the materials PP nature, PP beta 0.1, and PP beta 1, different strength plateaus are found for the dimensionless joining paths of 0.5 to 0.95. The plateau for PP nature indicates values above 35 MPa, while the plateau for PP beta 0.1 is about 30 MPa. If the amount of beta nucleating agent is increased, the short-term tensile strength is further reduced to 28 MPa, in line with results already shown in [22]. For the materials PP carbon black 0.1 and PP carbon black 1, a strength plateau is only seen for a dimensionless joining path of 0.75 and values above. This plateau is in the range of 36 MPa. Table 3 shows mean value of short- and long-term tensile strength of the welded PP samples for different dimensionless joining paths.

### 3.3. Relation between Short- and Long-Term Tensile Strength

Tüchert shows in [24] that the short-term tensile strength is not directly correlated to the long-term tensile strength. Thus, very good short-term tensile strength does not necessarily lead to excellent long-term tensile strength. Figure 11 shows the long-term tensile strength results of the materials investigated with the dimensionless joining paths from 0.3 to 0.95. To investigate the long-term tensile strength, in contrast to the short-term tensile strength, a creep tensile test according to DIN EN-12814-3 is used, with time in the y-axis used as a reference.

Concerning the materials PP nature and PP carbon black 0.1, the highest long-term tensile strength was achieved for the dimensionless joining path of 0.5. Thus, a load could be applied for about 2500 h in this case. These high values are thought to be imposed by the structural phases being present in each case. It can be deduced that a predominant α-modification favors long-term strength. With a dimensionless joining path of 0.3, most materials, except the modifications including beta-additives, show very poor mechanical long-term properties. For the dimensionless joining paths ranging from 0.75 to 0.95, the materials PP carbon black 0.1 and PP nature are characterized by plateaus at a very similar level. An average value of 350 h can be achieved for the long-term tensile strength of both materials. The material PP carbon black 1 shows an average long-term tensile strength of about 500 h starting from a dimensionless joining path of 0.5.

Obviously, the crystal modification is a key to assessing the different long-term tensile strength. Here, the dominating α-modification for PP nature and PP carbon black 0.1 is likely to be the main reason for the extraordinarily high long-term tensile strength at the dimensionless joining path of 0.5. The subsequent addition of carbon black (up to the higher weight percentage of 1%) reduced this effect, so that eventually the long-term joint was weakened by the very high content of carbon black. As is well-known, the high carbon black content leads to a higher fraction of the beta modification, which is therefore unfavorable for the long-term tensile strength. Preliminary tests have shown that there is no correlation between the different crystallinities and the joining paths.

Of all the materials, PP beta 1 shows superior long-term tensile properties with respect to all process parameter applied. The average number of operating hours achieved for the long-term tensile properties was 2000 h. PP beta 0.1 shows the same material behaviour as PP beta 1 for dimensionless joining paths of 0.3 and 0.5, i.e., 2000 operating hours were possible on average for these material conditions. The curve then flattens out so that a mean operating life of 1700 h was obtained for a dimensionless joining path of 0.75 and an average operating life of only 690 h was achieved for the dimensionless joining path of 0.95. These findings are in line with previous studies [24]. In this case, the high fraction of the beta phase favors the long-term tensile properties. This is in clear contrast to the considerations detailed above for the condition characterized by a large content of carbon black. Further analysis of the complex interplay of the prevailing phases cannot be deduced from literature and was out of the scope of the present study. Future studies including analysis of local deformation phenomena will need to shed light on the underlying elementary mechanism. However, the findings now presented should be carefully considered for the respective application, i.e., nucleation material should be only used if the final requirements can be met.

Comparing the short-term tensile strength (c.f. Figure 10) of the materials with the long-term tensile strength (c.f. Figure 11), it is apparent that the poor short-term tensile strength of the PP nature, PP carbon black 0.1, and PP carbon black 1 with a dimensionless joining path up to 0.5 also translate into low long-term tensile strength. For the good short-term tensile strengths from a dimensionless joining path of 0.75 and 0.95, long-term tensile strength in the range of 350 h to 500 h was achieved. Exceptionally high long-term tensile properties in the range of 2500 h were obtained for the process point of 0.5 for the materials PP carbon black 0.1 and PP nature.

Discussing these results in the context of Tüchert’s findings and statements [25], some differences are obvious. In the present study, it is shown that for good short-term tensile strength in the range of the dimensionless joining paths of 0.75 and 0.95, at least average long-term tensile strength (based on the comparison of the values achieved in the case of the beta nucleated samples) are attained for these materials. With regard to the dimensionless joining paths of 0.5, superior long-term tensile strength are revealed in parallel to high short-term strength. Clearly the beta-nucleated materials differ from the other material conditions tested. Very high long-term tensile strength are shown for the dimensionless joining paths from 0.5 to 0.95. With regard to the short-term tensile strength, it is found that high short-term strength can be transferred to high long-term tensile strength. For the dimensionless joining path of 0.3, however, it is shown that very good long-term tensile properties were achieved even if the short-term tensile properties of these materials are very low. This clearly indicates that Tüchert’s statement cannot be transferred to every material condition, i.e., poor short-term tensile strength can still result in high long-term tensile properties.

### 3.4. Residual Stresses

For the interpretation of the results regarding residual stresses, it has to be kept in mind that the analysis of the residual stresses only provides reliable data at a measuring depth range of 10 to 600 µm. The results in the direct vicinity of the surface and below the maximum depth of 600 µm are not reliable, see [21] for details. For the measurement of each welded sample, the base material was also considered in parallel to check whether the determined stresses of the welded samples are still compatible with the base material. In addition, this comparative measurement was used to validate whether compressive stresses occur in the base material (non-welded area) and tensile stresses occur in the weld, as highlighted in [26]. For each dimensionless joining path, only one sample was measured. Some preliminary tests show that the measurement of different samples with the same dimensionless joining path has a good repeatability.

The measurement results of the residual stress for PP carbon black 0.1 with different joining paths are shown in Figure 12a. The material PP nature was also measured for comparison. It is seen that the PP nature is characterized by compressive stresses with the value of around −2 MPa close to the surface of the sample. For the samples with the dimensionless joining paths of 0.3, 0.5, and 0.95, the tensile residual stresses appear close to the surface of the sample, whereas for the joining path of 0.75, the sample is characterized by compressive stress with a value of −11 MPa and the stress is then switched from compressive to tensile stress at a depth of 100 μm. Furthermore, it is shown that the highest residual stress level for the dimensionless joining path of 0.95 prevails at a depth of 400 µm. According to the results shown in Figure 12a, it can be implied that the curves cannot be assigned proportionally to the dimensionless joining paths due to the complex forming mechanism at the microstructural level and the resulting scattering effect. This is directly indicated by the fact that the curve for the dimensionless welding path of 0.75 is below the other curves. It is well known that compressive residual stresses close to the surface of the sample are able to enhance the long-term tensile strength, whereas the tensile residual stresses negatively affect the mechanical performance. However, as shown in Figure 11, for the joining path of 0.75 the compressive residual stresses close to the sample surface are found, however, poor long-term tensile strength appears. It can be implied that the microstructural phases determine the macroscopic mechanical performance and the residual stress has a negligible influence for the case with additive of carbon black 0.1%.

Figure 12b shows the residual stress results of the material PP carbon black 1 for the direction X perpendicular to the joint. It is seen that all samples are characterized by compressive residual stresses within the range of 1–4 MPa. As the depth increases, the residual stresses are decreased. In comparison to PP carbon black 0.1, it can be seen that the residual stress curve flattens rapidly, even at relatively low depth. The absolute values and trends seen for different samples are very close to each other, so that a differentiation with regard to the dimensionless joining path is not possible. The results found for the base material start in the compression range and touch the tensile range only very late, i.e., at a depth of 700 µm. The use of an increased amount of carbon black eventually promotes a decrease of the residual stresses independently of the dimensionless joining path (in the range of 0.3 to 0.95).

The results of the residual stress of the material PP beta 0.1 are shown in Figure 12c. It can be seen that, due to the beta nucleating agent, the residual stress values begin in the compression range (−2 to −3 MPa close to the sample surface), which contradicts literature values presented in [26]. In the range between 100 µm and 400 µm, it can be seen that the values for the two highest dimensionless joining paths (0.75 and 0.95) are above those of the other curves. The order of the other two curves is reversed, so that the small non-dimensional joining path of 0.3 lies above the curve of the dimensionless joining path of 0.5. An assignment to the dimensionless joining paths is therefore only possible here for the joining paths of 0.75 and 0.95. After the depth, they are switched into the tensile strength, which contributes to the fact that the residual stresses are reduced as the joining path increases.

Figure 12d shows the residual stress values of the material PP beta 1. Again, the curves are very similar, so that a differentiation with regard to the joining path is not possible. This is close to the case of PP carbon black 1. It can be concluded that, the higher the percentage of the additions, the greater the influence on residual stress compared with the pure material. The influence of joining path on residual stresses is small. The residual stress curves of almost all material conditions examined are below 0 MPa in the range of 100 to 200 µm, indicating that an increase in the beta content has led to a shift in the sign of the residual stresses from tensile to compressive stresses. This only applies to the area directly below the surface. After a drilling depth of 200 µm, the curves of the dimensionless joining paths cross the X-axis again and are thus characterized by tensile stresses. These tensile stresses increase up to a maximum of 2 MPa. A similar course as for the material PP beta 0.1 was found. The process conditions of the dimensionless joining path of 0.3 shows poor mechanical properties, whereas starting from a joining path of 0.5 the short-term tensile strength of the welded joint is about 28 MPa.

Due to the closely spaced curves of all examination points of the residual stresses, it is not possible to differentiate between the service life of the long-term tensile test. Thus, it is not possible to draw conclusions about the long-term tensile strength from the residual stress values.

In the context of process-structure drawing, there is as yet no knowledge of the extent to which residual stresses influence short- or long-term tensile strength. Only on the basis of these investigations carried out here does it become clear that the influence of residual stresses is negligible for the materials selected and the process settings. Thus, the residual stresses have only a minor influence on the long-term as well as the short-term tensile strength.

## 4. Conclusions

In the present study, polypropylene (PP) compounds with different additives (beta-nucleating and carbon black agents) were fabricated by the injection molding technology. These samples were joined by using hot plate welding. This work is chiefly concerned with the analysis of the effect of the additives and dimensionless joining path on the short- and long-term tensile strength, morphology, and residual stresses of the welded PP samples. The residual stresses were measured through the hole drilling method (HDM), taking into account the effect of thermal fluctuations and viscoelastic phenomena.

With regard to the long-term tensile properties, it was shown that the compound performance is decisively influenced by the addition of nucleating agents. It was found that, by using the beta-nucleating agent, outstanding long-term tensile properties are achieved as compared to untreated PP nature or PP sample with carbon black added. However, with regard to the short-term tensile strength, the beta-nucleating agent has a negative influence. After adding the agent, respective values are lower than in the case of unreinforced and the carbon-black polypropylene, respectively.

If residual stresses are considered in the direction of tension, it was found that these are homogenized by the increased use of the nucleating agent and shifted into the compression range. In this work, absolute values of residual stress are relatively low in all conditions. Linking residual stresses with long-term tensile strength is challenging but very significant as it directly affects the lifetime of the component and structures. This topic will be the covered in the follow-up work.

## Figures and Tables

**Figure 1 polymers-13-02965-f001:**
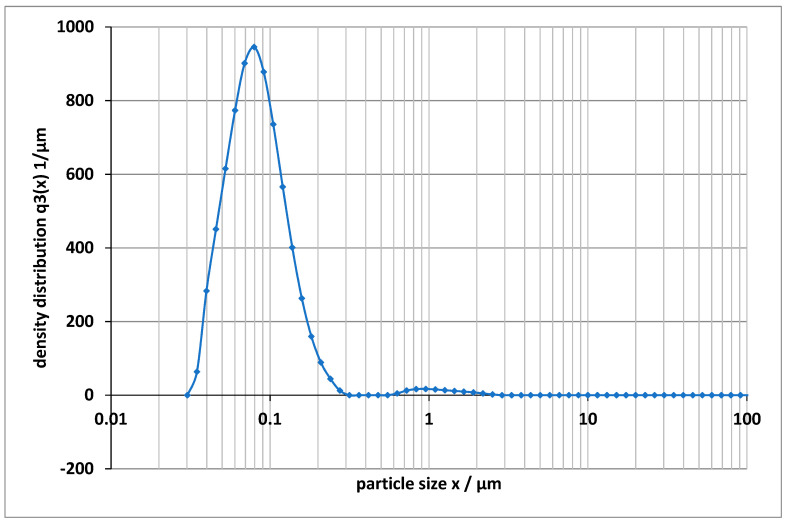
Particle size distribution of carbon black added to the polypropylene sample.

**Figure 2 polymers-13-02965-f002:**
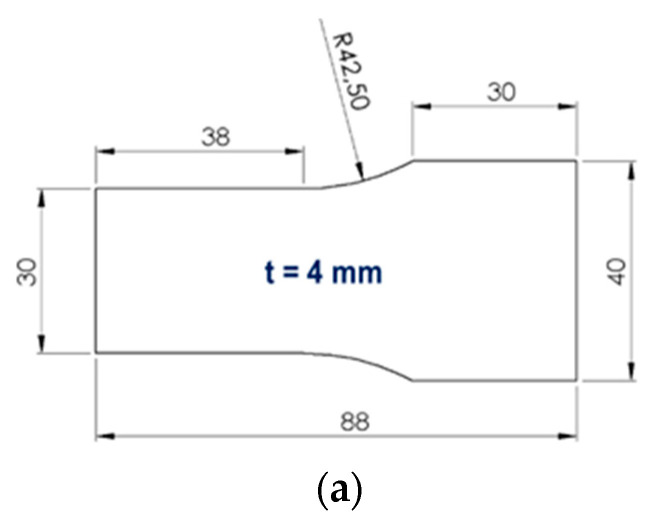
(**a**) Dimensions of the test sample to be welded (in mm), (**b**) sketch of joining path.

**Figure 3 polymers-13-02965-f003:**
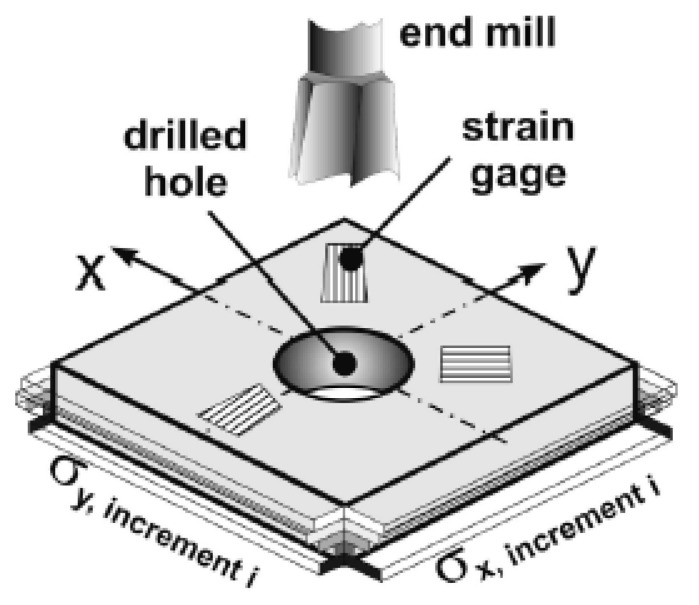
Schematic diagram highlighting the principle of the HDM, recompiled from [22].

**Figure 4 polymers-13-02965-f004:**
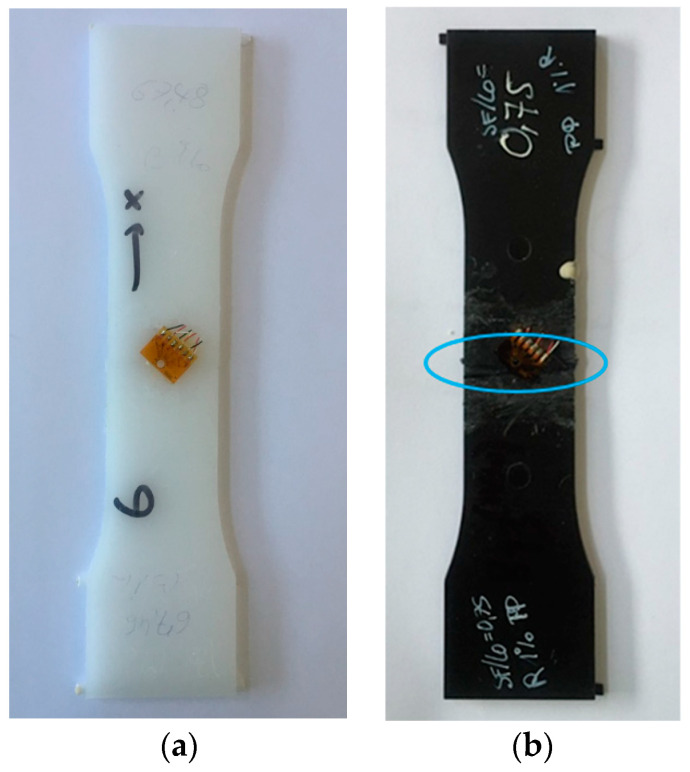
(**a**) Most relevant direction of the residual stress measurement and (**b**) photograph of a welded sample including an attached strain gauge and marking of the area considered in the following investigations (blue).

**Figure 5 polymers-13-02965-f005:**
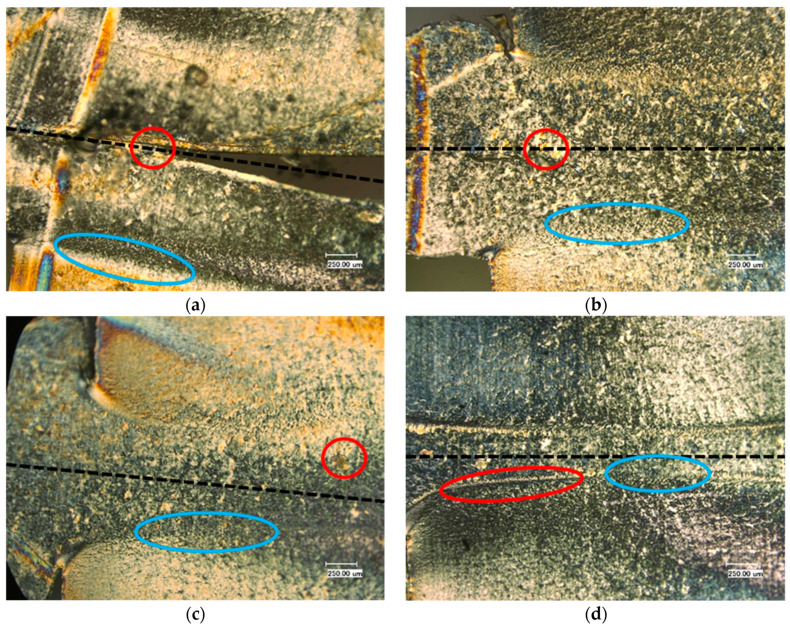
Microstructures of the welded PP nature with dimensionless joining path (**a**): 0.3; (**b**): 0.5; (**c**): 0.75; (**d**): 0.95; (**e**): WAXS measurement results of the welded PP nature for different joining paths. ● α-structure ● β-structure.

**Figure 6 polymers-13-02965-f006:**
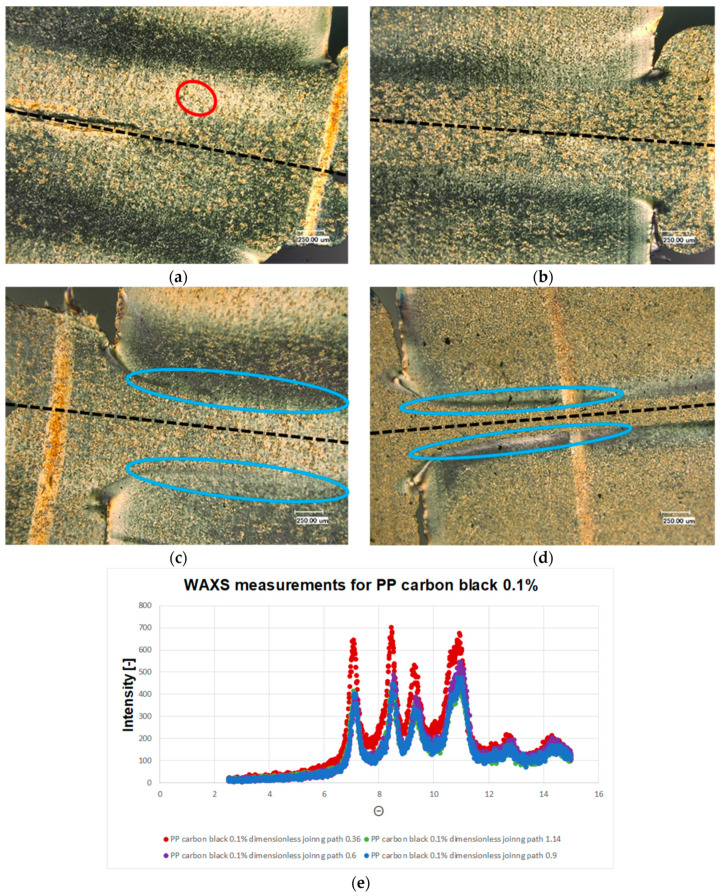
Microstructures of the welded PP carbon balck 0.1 with dimensionless joining path (**a**): 0.3; (**b**): 0.5; (**c**): 0.75; (**d**): 0.95; (**e**): WAXS measurement results of the welded PP carbon balck 0.1 for different joining paths. ● α-structure ● β-structure.

**Figure 7 polymers-13-02965-f007:**
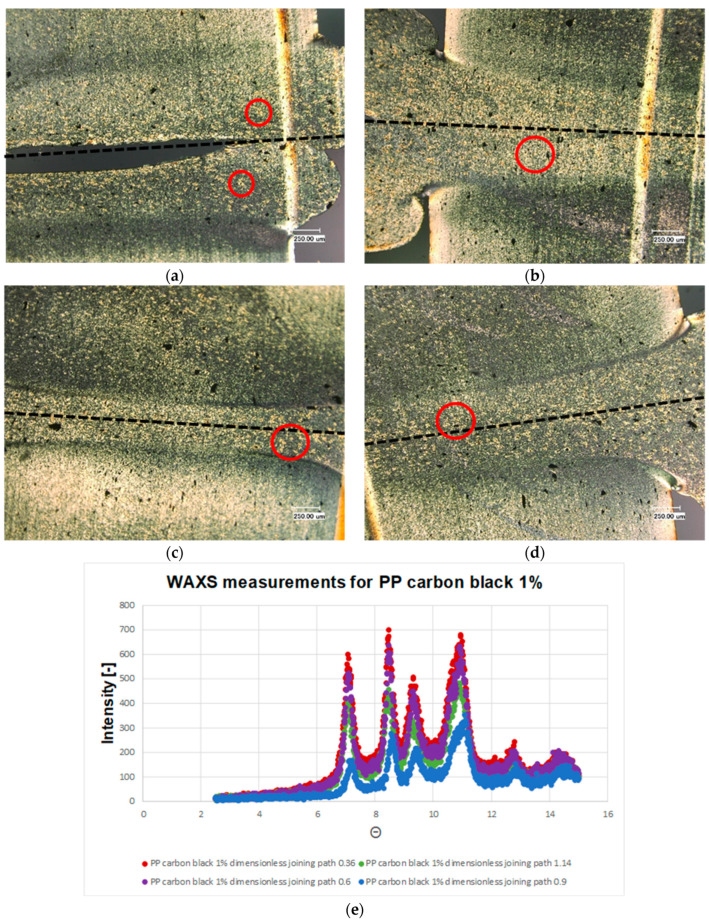
Microstructures of the welded PP carbon balck 1 with dimensionless joining path (**a**): 0.3; (**b**): 0.5; (**c**): 0.75; (**d**): 0.95; (**e**): WAXS measurement results of the welded PP carbon balck 1 for different joining paths. ● α-structure ● β-structure.

**Figure 8 polymers-13-02965-f008:**
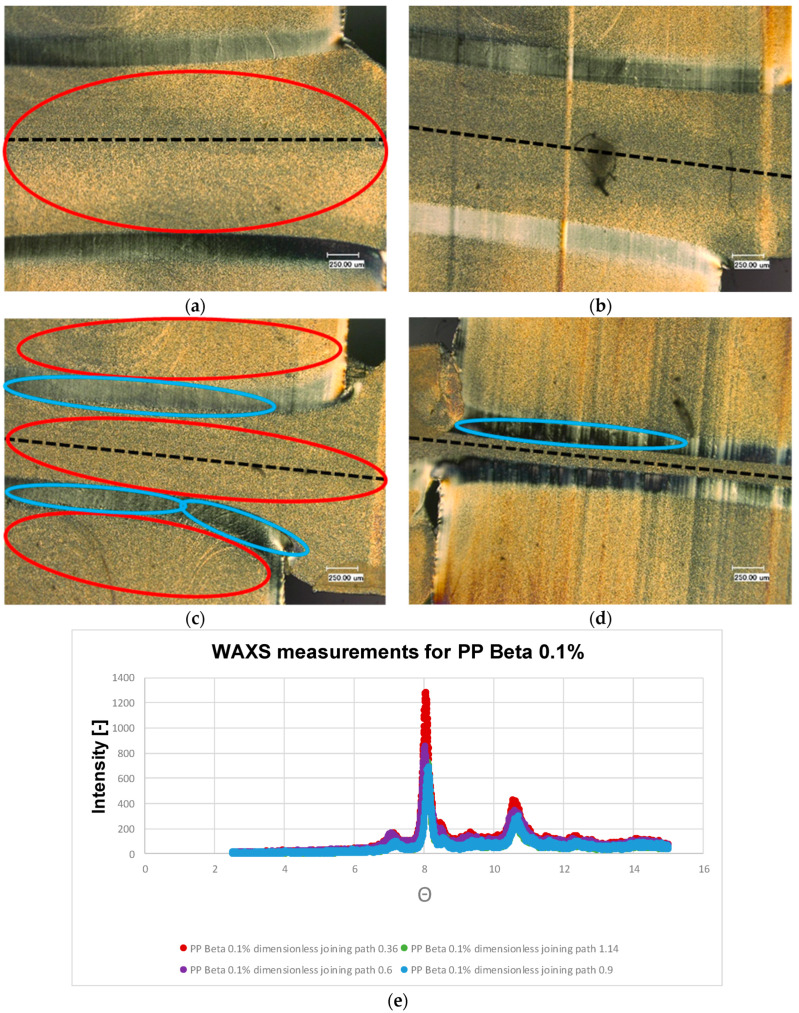
Microstructures of the welded PP beta 0.1 with dimensionless joining path (**a**): 0.3; (**b**): 0.5; (**c**): 0.75; (**d**): 0.95; (**e**): WAXS measurement results of the welded PP beta 0.1 for different joining paths. ● α-structure ● β-structure.

**Figure 9 polymers-13-02965-f009:**
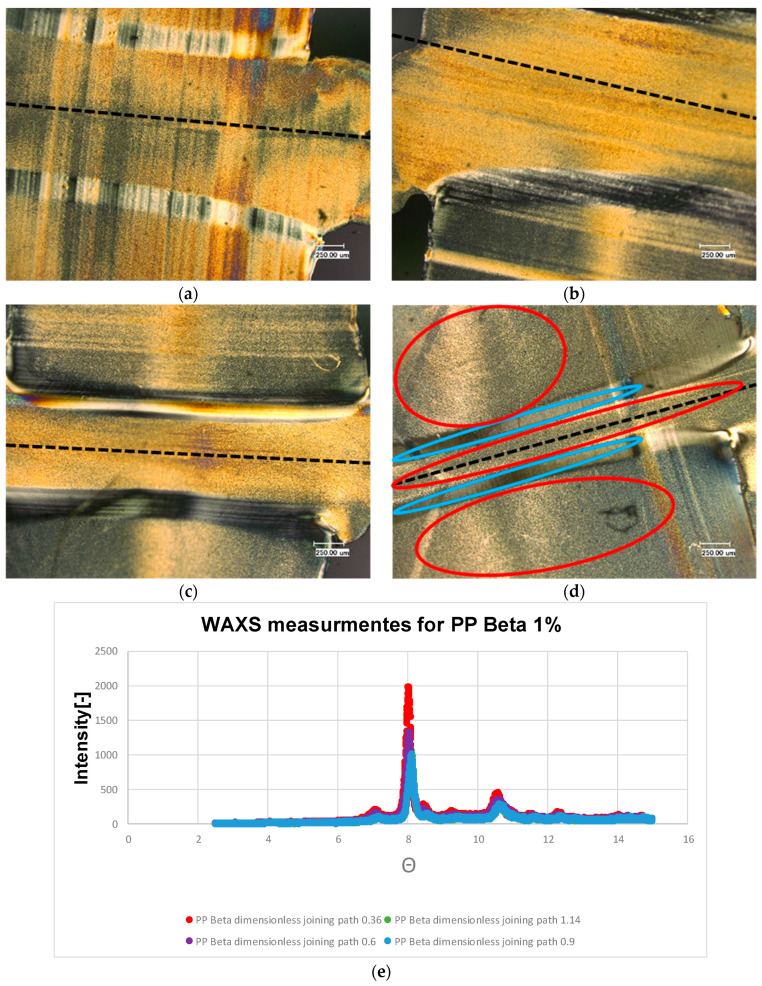
Microstructures of the welded PP beta 1 with dimensionless joining path (**a**): 0.3; (**b**): 0.5; (**c**): 0.75; (**d**): 0.95; (**e**): WAXS measurement results of the welded PP beta 1 for different joining paths. ● α-structure ● β-structure.

**Figure 10 polymers-13-02965-f010:**
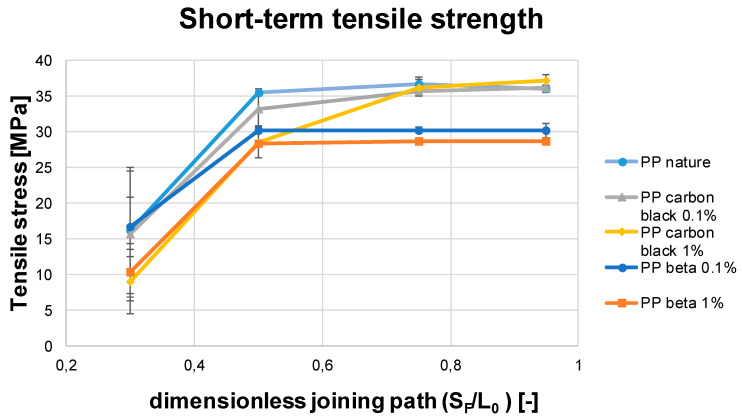
Short-term tensile strength of the welded PP samples for different dimensionless joining paths.

**Figure 11 polymers-13-02965-f011:**
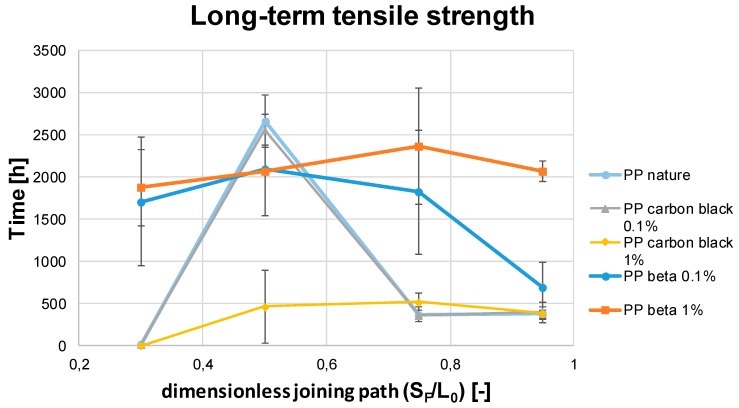
Long-term tensile strength of the welded PP samples for different dimensionless joining paths.

**Figure 12 polymers-13-02965-f012:**
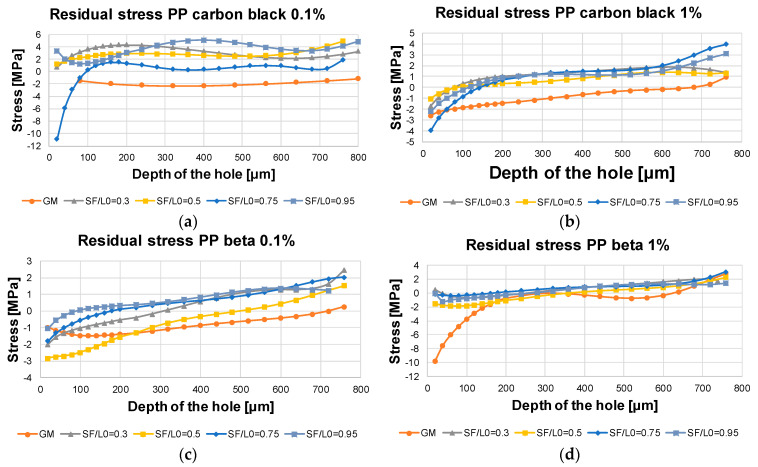
Residual stress results of the welded PP samples measured through hole drilling method (**a**) PP carbon black 0.1%, (**b**) PP carbon black 1%, (**c**) PP beta 0.1%, (**d**) PP beta 1%.

**Table 1 polymers-13-02965-t001:** List of materials used for analysis.

Material with Additive	Material Identification	Structure Phase
PP untreated (without additive)	PP nature	α
PP with 0.1 wt.% carbon black	PP carbon black 0.1	α and β (mainly α)
PP with 1 wt.% carbon black	PP carbon black 1	α and β (mainly α)
PP with 0.1 wt.% beta nucleating agents	PP beta 0.1	α and β (mainly β)
PP with 1 wt.% beta nucleating agents	PP beta 1	α and β (mainly β)

**Table 2 polymers-13-02965-t002:** Process settings in hot plate welding.

Heating Element Temperature = 220 °C	
Joining Path (sf) [mm]	Dimensionless Joining Path sf/Lo [-]
0.361	0.3
0.6	0.5
0.9	0.75
1.14	0.95

**Table 3 polymers-13-02965-t003:** Mean value of short- and long-term tensile strength of the welded PP samples for different dimensionless joining paths (unit: MPa, S, and L indicate short- and long-term tensile strength respectively).

	Material	Natur	Carbon Black 0.1	Carbon Black 1	Beta 0.1	Beta 1
Path						
S: 0.3		16.5	15.4	8.91	16.5	10.2
S: 0.5		35.5	33.2	28.3	30.1	28.3
S: 0.75		37.1	35.4	36.1	30.3	28.6
S: 0.95		36.0	36.1	37.1	30.4	28.7
L: 0.3		8.3	8.1	7.8	1663.1	1864.2
L: 0.5		2650.1	2540.4	456.9	2101.8	2065.2
L: 0.75		365.5	356.3	511.7	1809.3	2357.7
L: 0.95		374.6	374.6	365.5	667.10	2056.1

## Data Availability

The data presented in this study are available on request from the corresponding author.

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
