# Peer review of "Effect of Nucleating Additives on Short- and Long-Term Tensile Strength and Residual Stresses of Welded Polypropylene Samples"

_polymers, 2021, doi:10.3390/polym13172965_

Round 1

Reviewer 1 Report

This manuscript deals with the effect of nucleating agents on short- and long-term tensile strengths of welded polypropylene samples. The authors found some relationships between the short-time tensile strength and the long-term tensile properties in the samples added with a b-nucleating agent, which is interesting. Therefore, this manuscript can be accepted after minor revision. The authors are requested to reconsider the following minor points in the revision.

  • Describe the reason why the long-term tensile strength becomes the highest for the samples of PP Carbon Black 0.1 and PP Nature with a dimensionless joining path of 0.5?
  • Figures 3-6: If possible, add the WAXS data for the crystallinity in the supplementary section.
  • There are several syntax errors found in the text. For example:

87: retrain

92: Compunder

94: nanofilter

104: promanperties

341: tp

368: obcious

431: short-time

Author Response

  • Describe the reason why the long-term tensile strength becomes the highest for the samples of PP Carbon Black 0.1 and PP Nature with a dimensionless joining path of 0.5?

These high values are thought to be imposed by the structural phases being present in each case. It can be deduced, that a predominant alpha-modification favours long-term strength. The dominating alpha modification for the materials PP nature and PP carbon black 0.1 is expected to be the reason for the extraordinarily high long-term tensile strength at the dimensionless joining path of 0.5. All relevant information is provided in lines 368-385.

  • Figures 3-6: If possible, add the WAXS data for the crystallinity in the supplementary section.

The WAXS data has been added in the manuscript and some explanations to the data have also been provided, see lines 228-232, 254-255, 262-265.

  • There are several syntax errors found in the text. For example:

We appreciate the reviewer. These syntax errors have been revised.

87: retrain

92: Compunder

94: nanofilter

104: promanperties

341: tp

368: obcious

431: short-time

Reviewer 2 Report

  1. I think the abstract is too long for a technical paper. I suggest to shorten it by giving some numerical results in terms of %increase or decrease in the properties to make it more interesting to the reader.
  2. Lines 86-88: Please correct the language and spellings.
  3. Line 90: “adding nanocomposite….” Something is wrong here.
  4. Line 102: Grammatical mistake….”samples adding different additives”
  5. Line 104: “promanperties”…What is this?
  6. I guess the last paragraph of the Introduction part should be rewritten completely by carefully revising the language and spellings.
  7. The authors fail to highlight the need for the current study in the Introduction part. Please look into it.
  8. Lines 120 to 129 should be moved to the introduction part.
  9. The experimental procedure followed to form the composites with carbon black and beta nucleating agents are given very scantily. Should be elaborated properly.
  10. I did not see any discussion on dispersion of the nucleating agents. Please clarify.
  11. I guess, section 2.1 is too long. The authors should try to summarize this in a more proper manner.
  12. Error bars are missing for the residual stress plots. How many experiments were conducted for repeatability should be specified.
  13. The discussion part should be improved to explain the obtained results scientifically.
  14. It will be good to include a table summarizing the composites, which showed the best results for clarity purposes.

Author Response

Reviewers 2:

  1. I think the abstract is too long for a technical paper. I suggest to shorten it by giving some numerical results in terms of %increase or decrease in the properties to make it more interesting to the reader.

Thanks for this valuable suggestion. The abstract has been revised, see lines 26-42.

  1. Lines 86-88: Please correct the language and spellings.

We appreciate the reviewer. The language and spellings have been revised.  

  1. Line 90: “adding nanocomposite….” Something is wrong here.

Thanks for the comment. It has been changed to polypropylene-based polymers.

  1. Line 102: Grammatical mistake….” samples adding different additives”

We appreciate the reviewer. All mistakes have been revised.

  1. Line 104: “promanperties”…What is this?

Thanks. This was one typo and it has been replaced by properties.

  1. I guess the last paragraph of the Introduction part should be rewritten completely by carefully revising the language and spellings.

We are grateful for this valuable suggestion. That part has been revised, see lines 107-127.

  1. The authors fail to highlight the need for the current study in the Introduction part. Please look into it.

Thanks for the valuable comment. The relevant information has been added in lines 123-127;

  1. Lines 120 to 129 should be moved to the introduction part.

Thanks for this comment. It has been moved to the introduction part, see lines 93-105.

  1. The experimental procedure followed to form the composites with carbon black and beta nucleating agents are given very scantily. Should be elaborated properly.

Different additives including beta-nucleating and carbon black agents are added to the polypropylene material for improving mechanical performance. The mixing of the additives with the base material was done manually for preventing the molecular chains from breaking down. The properties of polypropylene depend on its molecular weight. Compounding in a twin screw was not performed upstream as it would lead to a shortening of the molecular chain. A particle size distribution (c.f. Figure 1) of the used carbon black was employed. This information has been added in lines 131-136.

  1. I did not see any discussion on dispersion of the nucleating agents. Please clarify.

The mixing of the additives with the base material was done manually, see lines 132-133.

  1. I guess, section 2.1 is too long. The authors should try to summarize this in a more proper manner.

Thanks for this valuable suggestion. The structure of the paper has been updated.

  1. Error bars are missing for the residual stress plots. How many experiments were conducted for repeatability should be specified.

For each dimensionless joining path, only one sample was measured. Some preliminary tests show that the measurement of different samples with the same dimensionless joining path has a good repeatability. This information can be found in lines 423-425.

  1. The discussion part should be improved to explain the obtained results scientifically.

Some discussions have been added, see lines 434-450.

  1. It will be good to include a table summarizing the composites, which showed the best results for clarity purposes.

Thanks for kind suggestion. Table 3 is provided in the manuscript.
